# Influence of Salinity on the Extracellular Enzymatic Activities of Marine Pelagic Fungi

**DOI:** 10.3390/jof10020152

**Published:** 2024-02-13

**Authors:** Katherine Salazar-Alekseyeva, Gerhard J. Herndl, Federico Baltar

**Affiliations:** 1Bio-Oceanography and Marine Biology Unit, Department of Functional and Evolutionary Ecology, University of Vienna, 1030 Vienna, Austria; gerhard.herndl@univie.ac.at; 2Bioprocess Engineering Group, Department of Agrotechnology and Food Sciences, Wageningen University and Research, 6708 WG Wageningen, The Netherlands; 3Department of Marine Microbiology and Biogeochemistry, Royal Netherlands Institute for Sea Research (NIOZ), University of Utrecht, 1790 AB Texel, The Netherlands

**Keywords:** Ascomycota, Basidiomycota, yeast, filamentous fungi, Michaelis–Menten kinetics, salinity effect

## Abstract

Even though fungi are ubiquitous in the biosphere, the ecological knowledge of marine fungi remains rather rudimentary. Also, little is known about their tolerance to salinity and how it influences their activities. Extracellular enzymatic activities (EEAs) are widely used to determine heterotrophic microbes’ enzymatic capabilities and substrate preferences. Five marine fungal species belonging to the most abundant pelagic phyla (Ascomycota and Basidiomycota) were grown under non-saline and saline conditions (0 g/L and 35 g/L, respectively). Due to their sensitivity and specificity, fluorogenic substrate analogues were used to determine hydrolytic activity on carbohydrates (*β*-glucosidase, *β*-xylosidase, and *N*-acetyl-*β*-D-glucosaminidase); peptides (leucine aminopeptidase and trypsin); lipids (lipase); organic phosphorus (alkaline phosphatase), and sulfur compounds (sulfatase). Afterwards, kinetic parameters such as maximum velocity (V_max_) and half-saturation constant (K_m_) were calculated. All fungal species investigated cleaved these substrates, but some species were more efficient than others. Moreover, most enzymatic activities were reduced in the saline medium, with some exceptions like sulfatase. In non-saline conditions, the average V_max_ ranged between 208.5 to 0.02 μmol/g biomass/h, and in saline conditions, 88.4 to 0.02 μmol/g biomass/h. The average K_m_ ranged between 1553.2 and 0.02 μM with no clear influence of salinity. Taken together, our results highlight a potential tolerance of marine fungi to freshwater conditions and indicate that changes in salinity (due to freshwater input or evaporation) might impact their enzymatic activities spectrum and, therefore, their contribution to the oceanic elemental cycles.

## 1. Introduction

In marine environments, fungi can live obligatorily or facultatively [1]. Obligate marine fungi can only grow in marine environments, whereas the facultative ones are species of terrestrial origin that can also grow in them [2]. Compared to their terrestrial and freshwater counterparts, marine fungi need to tolerate high salinity and limited substrate availability [3,4,5]. Therefore, marine fungi might have developed unique strategies to survive and thrive in this environment.

High salinity can cause osmotic and ionic stress to cells [6], hence, reduced survival [7]; however, fungi can perform several strategies to overcome this. One strategy is the “salt in” [8], which consists of equalizing internal and external salt concentrations, usually with KCl, but a special adaptation of the intracellular systems is needed [9,10]. Another strategy is the “compatible solute” [11], based on polyols, which are organic molecules that do not interfere with vital cellular functions [3,12]. A conserved pathway, known as the high-osmolarity glycerol (HOG), accumulates these polyols intracellularly to balance the osmotic pressure; hence, internal salt concentrations are preserved below toxic levels without any special adaptation of the intracellular systems [3,9,11,12,13,14]. Other strategies have also been reported in fungi, such as Na^+^/H^+^ antiporters that expel sodium ions from the cell and allow a turgor pressure within the extracellular environment [9]. Additionally, the cell wall is a crucial structure that protects them from environmental stresses [15], preserving the integrity and function of the stressed cells [11].

The biochemical versatility of fungi might be related not only to physiological adaptations, but also to their capability to use resources that suddenly become available [16]. Fungi as osmotrophs, synthesize different enzymes to break down high molecular weight polymers into smaller monomers that can then be absorbed by the cell wall [17,18,19]. The spectrum of secreted enzymes determines physiological needs [20], as well as adaptation to nutrition sources [18,21,22,23]. Thus, by releasing the necessary enzymes for the hydrolysis of polymers present in their respective environments [16], fungi are capable of changing between different lifestyles [18].

In land–sea transition zones, the role of marine fungi is similar to their terrestrial counterparts acting as decomposers, pathogens, and mutualistic symbionts [3,24]. The contribution of these fungi is unknown, but they might be involved in the recycling of organic carbon, nitrogen, and phosphorus [25,26]. However, the role of fungi in the open ocean is less known. Recent studies suggest that open-water fungi play an active role in marine biogeochemical cycles [27] and that they degrade a variety of carbohydrates [28]. Moreover, Gutiérrez, Pantoja [29] suggested that fungi are significant components of the marine microbial community involved in the breakdown of organic matter.

In the ocean, microbial enzymes are crucial for processing organic matter [30,31]. Since the enzymatic capabilities and substrate preferences differ between organisms [25], the study of extracellular enzymatic activities (EEAs) can be used as a functional trait to investigate potential functional diversity [32]. Fungi are also a group present in marine environments [29,33,34,35,36], but limited information is available on their ecological role and what factors affect their activities. Here, we used five marine fungi species, two belonging to the phylum Ascomycota and three to Basidiomycota, the two dominant pelagic fungal phyla [27,37,38], to study their enzymatic capabilities. As the number of substrates used can be suggestive of the functional metabolic potential and diversity [39], we used eight substrates as representatives of the main molecules found in the oceans [40,41,42]. Due to their sensitivity and specificity [43], we used fluorogenic substrate analogues widely used in aquatic environments [26,30,31,44,45]. Furthermore, for a greater understanding of the factors potentially influencing their activities and response to changing environmental conditions, we investigated the effect of salinity, as this has been reported as an important environmental variable influencing the enzymatic activities of marine bacteria [30].

The desalination of the oceans, also known as ocean freshening, is a freshwater input from ice melting, riverine, and precipitation that exceeds the evaporation, hence influencing the salinity of the oceans [21,46,47,48,49]. Moreover, this freshwater input can intensify stratification, which reduces the vertical mixing of the water column [46,47,50]. Drastic salinity changes lead to osmotic and ionic stress on the organism [3]. Even though most marine fungi are adapted to tolerate high salinity [1], they might not be halophilic [3]. Thus, how marine fungi might be influenced by salinity changes remains enigmatic.

## 2. Materials and Methods

### 2.1. Culture of Fungal Species

Strains of *Blastobotrys parvus* (HA 1620), *Metschnikowia australis* (HA 635), *Rhodotorula sphaerocarpa* (HB 738), and *Sakaguchia dacryoidea* (HB 877) were obtained from the Austrian Center of Biological Resources (ACBR). These species were originally isolated from the Antarctic Ocean at salinities of 34.2 [51], 32.5 to 36.4 [52], N/A [53], and 34.7 [54] g/L, respectively. The species *Rhodotorula mucilaginosa* was isolated from the Atlantic Ocean during the Poseidon Cruise in March 2019 at a salinity of 36.9 g/L. These pure isolates were maintained on yeast malt extract agar [55,56] at room temperature and were renewed monthly. The mentioned species were chosen as representatives of the most abundant pelagic phyla reported (Ascomycota dominate, Basidiomycota follow) [27,37,38]. 

Two media were prepared, both containing 2 g/L of glucose, malt extract, peptone, and yeast extract. To compare the effects of salinity, 35 g/L of artificial sea salts (S9883 Sigma-Aldrich, Vienna, Austria) were added to one medium to represent the average salinity of the global ocean. To the other medium, no salts were added (0 g/L) to represent freshwater. For both media, the pH was adjusted to 8.0. Both media were autoclaved, and afterward, 0.5 g/L of chloramphenicol was added to each medium to avoid bacterial contamination that could lead to bias in the enzymatic activities. In a vertical laminar airflow cabinet (Steril Bio Ban 72), an arbitrary amount of the pure isolates cultured on yeast malt extract agar for one week was taken with a sterile loop and diluted in artificial seawater (35 g/L sea salts S9883 Sigma-Aldrich) to achieve an OD_660_ ≈ 1 [57]. The optical density at 660 nm wavelength (OD_660_) was used as it minimizes the effect of the variation in cell sizes [58]. This was measured with a UV-1800 Shimadzu spectrophotometer, and 1 mL of the diluted fungal culture was added to 100 mL of autoclaved medium. Afterward, 150 mL of this medium containing fungi were filled into Schott bottles and incubated in triplicate at 5 °C on a rotary shaker (Jeio Tech ISS-7100 Incubated Shaker, Daejeon, South Chungcheong, Republic of Korea). Approximately 500 μL of the liquid culture was used to track the daily growth of the cultures via OD_660,_ and once the exponential phase was reached, samples with similar OD_660_ values were chosen in triplicates for further analyses (EEAs and biomass).

### 2.2. Measuring Extracellular Enzymatic Activity and Fungal Biomass Determination

The hydrolysis of the fluorogenic substrate analogues, such as 4-methylumbelliferyl *β*-D-glucopyranoside (M3633 Sigma-Aldrich), 4-methylumbelliferyl *β*-D-xylopyranoside (M7008 Sigma-Aldrich), and 4-methylumbelliferyl *N*-acetyl-*β*-D-glucosaminide (M2133 Sigma-Aldrich) was used to determine the potential activity of the enzymes *β*-glucosidase (BGL), *β*-xylosidase (BXY), and *N*-acetyl-*β*-D-glucosaminidase (NAG) respectively (Table 1). As these enzymes are capable of hydrolyzing part of the cellulose [59], chitin, and xylan [60,61,62,63], respectively, they were used as potential indicators of carbohydrate cleavage. For the potential enzymatic activity of lipase (OLE), alkaline phosphatase (APA), and sulfatase (SUL), the fluorogenic substrate analogues 4-methylumbelliferyl-oleate (75164 Sigma-Aldrich), 4-methylumbelliferyl phosphate (M8883 Sigma-Aldrich), and 4-methylumbelliferyl sulfate potassium salt (M7133 Sigma-Aldrich), respectively were used. OLE mediates the cleavage of fatty acids [64,65], whereas APA and SUL are indicative of the cleavage of phosphate and sulfate esters in molecules [45], respectively. The hydrolysis of N-succinyl-Ala-Ala-Pro-Phe-7-amido-4-methylcoumarin (L2145 Sigma-Aldrich) and t-butyloxycarbonyl-L-phenylalanyl-L-seryl-L-arginine-7-amido-4-methylcoumarin (3107-v PeptaNova) was used to identify the potential enzymatic activity of leucine aminopeptidase (LAP), and trypsin (TRY), both involved in the cleavage of proteins and peptides [45,66]. Finally, the fluorophores methylcoumaryl amide (MCA) (A9891 Sigma-Aldrich) and methylumbelliferone (MUF) (M1381 Sigma-Aldrich) were used to standardize the hydrolytic activity.

For the enzymatic assays, sterile 96-well microplates with F bottom and low protein binding (XT64.1, Carl Roth Karlsruhe, Baden-Wurtemberg, Germany) were used. The protocol was based on Hoppe [45] and Salazar Alekseyeva, Herndl, and Baltar [67]. The MCA fluorophore was dissolved in 2-methoxyethanol to a final concentration of 100 μM, 50 μM, 10 μM, and 1 μM, whereas MUF was diluted to a final concentration of 2000 μM, 1000 μM, 100 μM, and 50 μM. Both MCA and MUF served as fluorescent standards to normalize the fluorescence obtained in the incubations of the fluorogenic substrates. The liquid culture that contained the fungi that reached the exponential phase was used in biological triplicates. As standards, 15 μL of the respective fluorophore was added to 285 μL of the mentioned liquid culture, and as blanks, 300 μL of only this liquid culture was added. For the determination of the enzymatic activity, 30 μL of the respective fluorogenic substrate was added to 270 μL of liquid culture. This was serially diluted with the liquid culture to obtain 12 final substrate concentrations ranging from 1000 to 0.5 μM, except trypsin, which ranged from 500 to 0.2 μM. The volume was completed with an additional 150 μL of liquid culture, all also in biological triplicates. The emitted fluorescence was measured with FluoroLog^®^ Horiba at an excitation wavelength of 365 nm and an emission wavelength of 445 nm. An initial measurement (T0) was performed, followed by hourly measurements (T_1_, T_2_, and T_3_) over 3 h. The microplates were incubated in the dark at 5 °C between measurements.

To determine fungal biomass, 40 mL of the individual fungal cultures that reached the exponential phase were filtered gently onto pre-weighed and combusted (450 °C for 6 h) Whatman GF/F filters (WHA1825047 Sigma-Aldrich, 47 mm diameter). The filters with the collected fungal biomass were dried at 80 °C for 3 d. Thereafter, the samples were weighed again, and the biomass was determined.

To determine the cell-specific biomass, 1.5 mL of sample that reached the exponential phase were filtered with a pluriStrainer Mini (43-10040-50 pluriSelect, 40 µm mesh size) to obtain a single-cell suspension. The sample was fixed in the dark with 0.5% (final concentration) glutaraldehyde for 10 minutes and subsequently frozen at −80 °C until further processing. Due to the multicellular structures of the filamentous species *Blastobotrys parvus*, its cell abundance could not be determined. For the other species, depending on the optical density, 10 to 40 μL of the sample was diluted with TE to obtain a final volume of 500 μL which was later stained with 5 μL SYBR^®^ Green 100× (S9430, Sigma-Aldrich). The cell abundance was determined using a BD Accuri™ C6 Plus Flow Cytometry set at “Run with limits” of 10,000 events and “Medium” and the cell abundance was obtained with the BD Accuri C6 Software version 264.

### 2.3. Determination of the Enzyme Kinetics

The increased fluorescence over time in the samples with the fluorogenic substrate added was converted into a hydrolysis rate (μmol L^−1^ h^−1^) using the standard calibration with MCA and MUF. The obtained hydrolysis rates were fitted directly to the Michaelis–Menten equation using nonlinear least-squares regression analysis with R software version 4.3.2. [68]. The kinetic parameters V_max_ and K_m_ were calculated. V_max_ represents the maximum reaction value, which is independent of the substrate concentration, whereas K_m_ is the enzyme’s affinity to the substrate [69]. For the biomass-specific activity, the V_max_ was normalized to the dry weight (μmol/g biomass/h). The cell-specific activity was obtained by normalizing the hydrolysis rate to the cell abundance and is given in amol/cell/h.

### 2.4. Statistical Analysis

Shapiro–Wilk test was used to evaluate distribution. For the kinetic parameters V_max_ and K_m_, one-way Analysis of Variance (ANOVA) was used to determine whether the differences found between fungal species were significant. Tukey’s honestly significant difference (Tukey’s HSD) was performed to identify significance at the species level in the respective salinity after variances were tested (Levene’s test). 

## 3. Results

All the fungal strains used in this study were able to grow in non-saline (0 g/L) and saline (35 g/L) media (Figure 1) and hydrolyze all the fluorogenic substrates targeting carbohydrates (Figure 2), lipids, phosphorus, and sulfur moieties (Figure 3), and peptides (Figure 4). The results obtained suggested that the substrate concentrations needed to achieve half of the maximum velocity were variable among the different fungal strains (Figure 5). Generally, the extracellular enzymatic activities (EEAs) were higher in the non-saline than in the saline medium (Figure 6).

### 3.1. Carbohydrate Cleavage

#### 3.1.1. β-Glucosidase (BGL)

Compared to the other fungal species, *S. dacryoidea* exhibited a significantly higher V_max_ for BGL (*p* < 0.001; Figure 2(1). This was significantly higher at a salinity of 0 g/L (170.1 ± 47.7 μmol/g biomass/h and 9015.0 ± 3453.3 amol/cell/h) than at 35 g/L (10.7 ± 3.2 μmol/g biomass/h and 1853.4 ± 283.6 amol/cell/h) (*p* < 0.001). Even though the other fungal species exhibited a generally low BGL hydrolysis rate, this was significantly higher at a salinity of 0 g/L (*p* < 0.001), except for *R. mucilaginosa*. Interestingly, this was the only species that exhibited a higher BGL activity at a salinity of 35 g/L (0.07 ± 0.02 μmol/g biomass/h and 7.6 ± 2.4 amol/cell/h) compared to 0 g/L (0.02 ± 0.00 μmol/g biomass/h and 7.0 ± 1.3 amol/cell/h) (*p* < 0.001).

*Sakaguchia dacryoidea* exhibited a higher BGL activity and a higher K_m_ (767.3 ± 251.3 μM) at a salinity of 0 g/L than all the other fungal strains tested (*p* < 0.001; Figure 5A). The K_m_ values of the other Basidiomycota species (*R. mucilaginosa* and *R. sphaerocarpa*) were also higher in the non-saline than in the saline medium (*p* = 0.001). In contrast, there was no clear influence of salinity on the K_m_ of the Ascomycota strains *B. parvus* and *M. australis* (*p* = 0.2).

#### 3.1.2. β-Xylosidase (BXY)

For BXY, *S. dacryoidea* also exhibited a significantly higher V_max_ than the other fungal strains (*p* < 0.001; Figure 2(2)). The V_max_ of this species was significantly higher in the non-saline (9.3 ± 2.1 μmol/g biomass/h and 468.9 ± 114.8 amol/cell/h) than in the saline medium (2.1 ± 1.0 μmol/g biomass/h and 348.3 ± 124.4 amol/cell/h) (*p* = 0.01). Similarly, *R. sphaerocarpa* exhibited a significantly higher V_max_ at a salinity of 0 g/L (*p* < 0.001) than at 35 g/L. The two Ascomycota species *B. parvus* and *M. australis*, and the Basidiomycota species *R. mucilaginosa* showed low BXY activity.

*Blastobotrys parvus*, *R. mucilaginosa*, and *S. dacryoidea*, displayed a significantly higher K_m_ at a salinity of 35 g/L than at 0 g/L (*p*= 0.004; Figure 5B). *S. dacryoidea* exhibited the highest K_m_ value (1036.3 ± 397.9 μM) at 35 g/L of all the fungal strains tested, followed by *B. parvus* also at 35 g/L (631.8 ± 382.4 μM). In the non-saline medium, the K_m_ for *S. dacryoidea* was only about half (521.9 ± 165.7 μM) of the one determined in the saline medium. Interestingly, *M. australis* presented the lowest K_m_ in both media (10.9 ± 4.3 μM).

#### 3.1.3. *N*-Acetyl-β-D-glucosaminidase (NAG)

Even though all the fungal strains were capable of using the fluorogenic substrate for NAG, the overall activity rates were low (Figure 2(3). The V_max_ of *B. parvus* at a salinity of 0 g/L (2.4 ± 0.5 μmol/g biomass/h) was significantly higher than the V_max_ of the other fungal species (*p* < 0.001). Also, the species of the Basidiomycota phylum, *R. mucilaginosa*, *R. sphaerocarpa*, and *S. dacryoidea* exhibited a higher V_max_ in the non-saline medium (*p* = 0.001), whereas *M. australis* exhibited a higher V_max_ in the saline medium (*p* = 0.009).

In the non-saline medium, *B. parvus* exhibited the highest NAG activity and with 477.4 ± 150.6 μM also the highest K_m_ (Figure 5C), so it was significantly higher than the K_m_ for NAG of the other fungal strains (*p* = 0.004). The K_m_ of the other fungal strains varied between 46.7 and 0.01 μM.

### 3.2. Hydrolytic Activity on Organic Compounds Containing Lipids, Phosphorus and Sulfur Moieties

#### 3.2.1. Lipase (OLE)

When normalizing OLE activity to biomass, *B. parvus* and *S. dacryoidea* exhibited significantly higher V_max_ values than the other species (*p* < 0.001; Figure 3(1)). Nevertheless, when OLE activity was normalized to cell-specific activity, *S. dacryoidea* exhibited the highest V_max_ of all the species (*p* < 0.001), specially in the saline medium (13,159.9 ± 4080.3 amol/cell/h). *M. australis*, *R. mucilaginosa*, and *R. sphaerocarpa* were also capable of cleaving lipids, but at much lower rates ranging from 2.9 to 0.03 μmol/g biomass/h and 127.6 to 0.5 amol/cell/h, and there was not a clear pattern on the effect of salinity (*p* = 0.1).

Generally, in all fungal strains, the K_m_ was significantly higher in the saline than in the non-saline medium except in *R. sphaerocarpa* and *B. parvus* (*p* < 0.001; Figure 5D), where the K_m_ was significantly higher in the non-saline medium (*p* = 0.01) and was 478.3 ± 284.7 μM. In all the other fungal strains, the K_m_ varied between 172.8 and 0.6 μM in the non-saline medium and from 700.7 to 29.1 μM in the saline medium.

#### 3.2.2. Alkaline Phosphatase (APA)

Normalizing APA activity to biomass revealed that the V_max_ of *M. australis* in the non-saline medium was significantly higher than the V_max_ of the other species (*p* < 0.001; Figure 3(2)). When the V_max_ was normalized by the cell abundance, *S. dacryoidea* exhibited the highest V_max_ in the saline medium (757.2 ± 201.1 amol/cell/h) compared to the other fungal strains (*p* < 0.001), followed by *M. australis* with a V_max_ of 182.4 ± 43.1 amol/cell/h in the non-saline medium. Similarly, *R. mucilaginosa* and *R. sphaerocarpa* showed higher V_max_ in the non-saline medium (*p* < 0.001) than the other fungal strains except *M. australis*, whereas *B. parvus* exhibited a higher V_max_ in the saline medium (*p* = 0.02).

Under non-saline conditions, *R. mucilaginosa* exhibited a significantly lower V_max_ than the other fungal strains (*p* < 0.001; Figure 3(2)), while its K_m_ (506.2 ± 103.8 μM) was the highest (*p* < 0.001; Figure 5E). While in *M. australis* and *S. dacryoidea* the K_m_ was lower in saline conditions than in the other conditions (*p* < 0.001). The K_m_ values of *B. parvus* and *R. sphaerocarpa* were not significantly different between both conditions (*p* = 0.2).

#### 3.2.3. Sulfatase (SUL)

All fungal strains studied were capable of cleaving sulfate esters, however, at low rates (Figure 3(3)). All the species, except *R. sphaerocarpa*, showed a significantly higher V_max_ in saline than in the non-saline medium (*p* = 0.05). The highest V_max_ in the saline medium was detected in *S. dacryoidea* (0.8 ± 0.4 μmol/g biomass/h and 124.1 ± 45.9 amol/cell/h) while in *R. sphaerocarpa* the V_max_ was higher in the non-saline than in the other medium (*p* < 0.001).

However, as shown in Figure 5F, there was no clear effect of salinity on the K_m_ of sulfatase (*p* = 0.3). Significantly higher values were found for both salinities in *B. parvus* than in the other strains, with K_m_ values of 524.7 ± 220.6 μM in the non-saline and 493.3 ± 160.6 μM in the saline medium.

### 3.3. Proteins Cleavage

#### 3.3.1. Leucine Aminopeptidase (LAP)

For the species of the phylum Basidiomycota, *R. sphaerocarpa*, and *S. dacryoidea*, the V_max_ for LAP was significantly higher in the non-saline than in the saline medium (*p* = 0.004; Figure 4(1)). The V_max_ values of the Basidiomycota species ranged from 208.5 to 2.9 μmol/g biomass/h, and between 10,101.1 to 604.4 amol/cell/h.

The K_m_ was significantly higher in the Basidiomycota (except *R. mucilaginosa*) than in the Ascomycota strains (*p* < 0.001; Figure 5G). Compared to the other enzymes, the LAP K_m_ values were the highest. Even though there was no clear influence of the salinity on the K_m_ for all the species except *R. sphaerocarpa* (*p* = 0.5), *S. dacryoidea* exhibited high K_m_ values (1536.7 ± 832.8 μM in the non-saline and 1415.4 ± 615.3 μM in the saline medium).

#### 3.3.2. Trypsin (TRY)

Similar to LAP, the V_max_ values for TRY in all the fungal strains were significantly higher in the non-saline than in the saline medium (*p* < 0.001; Figure 4(2)). Comparing the biomass-specific TRY activity and the cell-specific activity, the overall pattern did not change. However, when calculating the V_max_ based on the biomass-specific TRY activity, *M. australis* exhibited a significantly higher V_max_ (3.2 ± 1.2 μmol/g biomass/h) in the non-saline medium (*p* < 0.001) than all the other strains, while the V_max_ calculated from the cell-specific activity, was higher in *S. dacryoidea* in the non-saline medium (51.8 ± 10.9 amol/cell/h; *p* < 0.001) than in all the other fungal strains. 

For all the fungal strains except *B. parvus*, K_m_ values were significantly higher in the saline medium (*p* < 0.001; Figure 5H). For this species, there was no clear influence of the salinity on the K_m_ (*p* = 1.0). The highest K_m_ detected corresponded to *M. australis* and *R. mucilaginosa* in the saline medium (471.7 ± 72.7 μM and 407.0 ± 136.0 μM, respectively).

## 4. Discussion

The use of fluorogenic substrate analogues allowed the determination of potential extracellular enzymatic activities of marine pelagic fungal isolates. The fungal strains studied, *B. parvus*, *M. australis*, *R. mucilaginosa*, *R. sphaerocarpa*, and *S. dacryoidea* exhibited a broad spectrum of extracellular enzymes in saline and non-saline media. They were able to enzymatically hydrolyze part of carbohydrates like cellulose, chitin, and xylan, and proteins such as peptide chains and leucine residues. Moreover, they expressed enzymes to hydrolytically cleave lipids, phosphate-containing compounds, and sulfate esters. Since these measurements were performed during the exponential phase, fungi were allowed to adapt to the new media. Finally, a short incubation period (3 h) and the addition of chloramphenicol avoided the potential growth of other microorganisms that could cause bias in these measurements.

### 4.1. Extracellular Enzymatic Activities of Pelagic Fungal Strains

#### 4.1.1. Release of Enzymes Hydrolytically Cleaving Carbohydrates

Cellulose is the most common natural polysaccharide [70], so its degradation is crucial in the carbon cycle [71]. Several microorganisms, specially fungi, perform cellulolytic activities with three major classes of enzymes such as endoglucanases, exoglucanases, and *β*-glucosidases [72,73,74,75,76]. The hydrolysis of cellulose has been reported by a wide variety of marine fungi [67,75,77,78,79,80,81,82,83,84], but their capability to decompose this complex compound depends on their enzymatic machinery [85]. Vaz, Rosa [86] and Gonçalves, Paço [87] showed that 76% and 68%, respectively, of the marine fungi they studied were able to release enzymes related to cellulolytic activity. However, compared to other cellulolytic enzymes, *β*-glucosidase hydrolyzes the products of those enzymes to glucose [88]. In the soil, a widely distributed species, *Aspergillus niger*, exhibited a V_max_ of 11.7 μmol/g/h, and other fungal species, had values between 14.8 and 6.7 μmol/g/h [89]. All our fungal isolates were able to hydrolytically cleave cellulose, with *S. dacryoidea* dominating this cellulolytic activity among all the strains tested (Figure 2(1)). Moreover, all the species exhibited K_m_ values > 97.0 μM, except the two species belonging to the genus *Rhodotorula* under saline conditions (Figure 5A). Thus, even though the studied species can use this type of cellulolytic enzyme, the K_m_ values obtained indicate that relatively high cellulose concentrations are preferred. In marine environments, cellulose is present in mangroves and associated with hemicellulose and lignin [90,91], but it is also present in some algal species [92]. As the organic matter of terrestrial origin is more available in coastal areas and less abundant in the open ocean [93,94,95], only specific microorganisms might be able to use cellulose [96] and dominate its decay in the open ocean.

After cellulose, chitin is the second most abundant polysaccharide [70,97,98,99] and is composed of monomers of *N*-acetyl-D-glucosamine (GlcNAc) linked by *β*-glycosidic bond [100,101,102,103]. Hexosaminidases are enzymes catalyzing the cleavage of these glycosidic bonds [97,104]. As chitin is abundant in marine environments [17], several marine fungal species are efficient chitin degraders [67,84,87,105,106], and together with bacteria, they are the main chitin degraders [102]. Fungi can also hydrolyze chitin, but unlike bacteria, fungi can use it as a building block for new chitin synthesis [107] or as a carbon and nitrogen source [102]. Nonetheless, dissimilarities between hyphae-like fungi and yeast chitinolytic enzymes have been reported [108]. The amount of chitinases released by fungi is related to their chitin content, so it is also dependent on the growth mode [109]. As hyphae-like fungal cell walls contain 10 to 20% of chitin [99], these fungi typically have 10 to 30 chitinases [110,111]. In contrast, yeast cell walls have a chitin content as low as 0.5 to 5% [112]. A hyphae-like fungi species, *Fusarium oxysporum*, exhibited a V_max_ of 20.7 μmol/g/h, and for its isoenzymes, 4.3 μmol/g/h [113]. *B. parvus*, also a filamentous fungus, had the highest chitinase V_max_ detected, under non-saline conditions (Figure 2(3)), and also presented the highest K_m_ (Figure 5C). In filamentous fungi, chitinolytic enzymes are important for cell wall regeneration and hyphae formation [114]. For the species not forming hyphae, their low K_m_ indicate a high affinity of the enzyme to the substrate at low concentrations [22]. Hence, all the studied species can use chitin, even when present at low concentrations, but hyphae-like species (*B. parvus*) might have an advantage over yeasts.

Due to xylan heterogeneity and complexity, several enzymes with diverse specificities are required to cleave it [115,116,117,118]. Thus, the xylanolytic activities are performed by xylanases, *β*-xylosidases, α-arabinosidases, α-glucuronidases, and esterases [118,119]. Widely distributed fungal species such as *Trichoderma viride* and *Emericella nidulans* exhibit *β*-xylosidase activity [120]. Biely, Vršanská, and Krátký [121] reported low *β*-xylosidase activity for *Cryptococcus albidus* but high xylanase activity, so 1,4 *β*-xylanase was highlighted as the main “xylan-attacking enzyme”. For the first enzyme, these authors obtained V_max_ values as low as 0.06 μmol/g/h. In the case of pelagic marine fungi, Salazar Alekseyeva, Herndl, and Baltar [67] also reported low *β*-xylosidase activity which is consistent with the present study, where the two species of the phylum Ascomycota, *B. parvus* and *M. australis*, and one species of Basidiomycota, *R. mucilaginosa*, also exhibited generally low *β*-xylosidase activity (Figure 2(2)). *R. sphaerocarpa* and *S. dacryoidea*, however, exhibited higher enzymatic activity in a non-saline than in the saline medium. As shown in Figure 5B, in the saline medium, *B. parvus*, *R. mucilaginosa*, and *S. dacryoidea* exhibited a high K_m_, indicative of a low affinity to the substrate [68] and inefficient organisms to obtain resources present at low concentrations [22]. In Antarctic environments, where the majority of the studied fungi were initially isolated [51,52,53,54], xylan is limited, but it can be found in the cell walls of green and red algae [122]. Thus, the low K_m_ or high affinity to the substrate of the only endemic species, *M. australis* [53,123], might be an advantage over the other species at the time of cleaving xylan.

#### 4.1.2. Release of Enzymes Cleaving Lipids, Phosphorus, and Sulfur Moieties

Lipase is the third largest enzyme group that can catalyze both hydrolysis and synthesis of esters and fatty acids [124,125]. These enzymes have been widely reported in marine fungi like *Hortaea werneckii*, *Trimmatostroma salinum* [126], *Rhodotorula glutinis* [127], *Yarrowia lipolytica* [128,129], *Leucosporidium scottii* [130], and in some Antarctic Ocean [131] and China Sea [132] strains. A species also included in the present study, *R. mucilaginosa* was suggested to be a promising fungal strain for lipase production [133], similar to Wang, Chi [132]. Contrary, we found that other species, such as *B. parvus* and *S. dacryoidea* had a higher OLE activity than *R. mucilaginosa* (Figure 3(1)), but the substrate dissociates easily from the enzyme (Figure 5F). 

Inorganic phosphate (Pi, PO_4_^3−^) is the preferred phosphorus source by microorganisms [134,135]. In surface waters, it is readily used by bacteria and phytoplankton, leading to low PO_4_^3−^ concentrations in vast areas of the global ocean [135,136,137,138]. To overcome this P-limitation, microorganisms release alkaline phosphatase to use a range of available dissolved organic phosphorus (DOP) [136,139,140]. This release might also be related to sporadic pulses of organic matter [141,142]. APA has been reported in marine fungi species like *Debaryomyces hansenii*, exhibiting a V_max_ of 0.83 μmol/g/h [143], and some species of Ascomycota and Basidiomycota phylum [67]. At a sporadic high organic matter concentration, a high K_m_ might be advantageous by allowing higher hydrolysis rates, for instance, for species like *B. parvus* (Figure 3(2) and Figure 5E). Nonetheless, a low K_m,_ such as for *S. dacryoidea*, might be more suitable to overcome the P-limitation most of the time when organic matter is low.

In contrast to terrestrial polysaccharides, numerous marine polysaccharides are highly sulfated [144,145,146]. For marine organisms, sulfated polysaccharides might be a physiological adaptation to the high ionic strength of their environment [146,147,148,149]. Nonetheless, microorganisms need to remove the sulfate groups to gain access to the carbohydrates [150,151]. The terrestrial fungus *Fusarium proliferatum* [152] and the marine fungus *Paradendryphiella salina* [84] were reported to assimilate sulfates from brown algae. In the present study, all the marine fungal species, except *R. sphaerocarpa*, exhibited a higher sulfatase activity in the saline medium (Figure 3(3)). The K_m_ was low for the species of the genus *Rhodotorula* (Figure 5F), but for the other examined species, the K_m_ was high. This might indicate an advantage for the species *R. mucilaginosa* and *R. sphaerocarpa*, as fewer enzymes and substrates would be needed to speed up this reaction in saline conditions, which would also allow the use of sulfated polysaccharides, common in marine environments.

#### 4.1.3. Release of Enzymes Hydrolytically Cleaving Proteins

Leucine aminopeptidase (LAP) is a ubiquitous enzyme involved in the hydrolysis of peptides [153,154,155]. In marine environments, the microbial production of LAP might indicate the use of nitrogen from suspended and sinking organic matter [30,44,156], so LAP is a key player in both carbon and nitrogen recycling [154]. Gutiérrez, Pantoja [29] associated an increase in LAP and BGL activities in surface waters with the availability of new organic matter during periods of high phytoplankton biomass production. Thus, microorganisms might respond to changing substrate availability by producing LAP [156]. In the present study, due to higher V_max_ values detected in the non-saline medium, it seems that the fungal strains produce less LAP in marine environments than in freshwater environments (Figure 4(1)). In the study of Caruso and Zaccone [30], changes in LAP activity of marine microbial communities were related to physical (salinity and temperature), chemical (nutrient composition), and microbiological variables. In our study, Basidiomycota species expressed a higher LAP activity than the Ascomycota species. As the substrate concentration needed to achieve half the maximum velocity is considerably high (Figure 5G), in pelagic environments where the organic matter is readily used within the water column [157] and decreases with depth [158], the activity of LAP by marine fungi might be preferably associated to environments with relatively high organic matter content.

Trypsin is an enzyme that cleaves proteins, specifically at the carboxyl end of lysine and arginine [159]. Comparable to leucine aminopeptidase, the production of this enzyme also depends on the nitrogen source [160,161], so it is important in the carbon and nitrogen cycles [162]. *Aspergillus ustus*, a deep-sea isolate from the Central Indian Basin, was a high producer of serine protease [163]. In the present study, we found that *R. mucilaginosa* had a high trypsin activity, but also other species, such as *S. dacryoidea*, specially in the non-saline medium (Figure 4(2)). Nonetheless, only *R. sphaerocarpa* presented relatively low K_m_ values (Figure 5H). As trypsin might be used under nitrogen deficiency to acquire this element [164], only a high affinity to the substrate might be beneficial to convert the substrate efficiently into products that can be used for this purpose [165].

### 4.2. Influence of Salinity on Extracellular Enzymatic Activity

Some fungi do not require salts to grow [6], but as reported by Jones and Jennings [166], salinity might not affect remarkably the growth of fungi in marine environments compared with freshwater and terrestrial environments. Similar results were found by Arfi, Chevret [167]. However, we found that even though salinity does not remarkably influence the growth of some fungi (Figure 1), it might exert an important effect on the kinetics of certain enzymes (Figure 2, Figure 3, Figure 4 and Figure 5).

In marine environments, Caruso and Zaccone [30] stated that salinity was an important variable affecting microbial enzymatic activities. In soil studies, a salinity increase leads to the exponential decrease of some microbial enzymatic activities such as alkaline phosphatase and *β*-glucosidase [168]. For the halotolerant species *Debaryomyces hansenii*, a high salinity resulted in low alkaline phosphatase activity [143]. In the present study, we also found that the majority of extracellular enzymatic activities were reduced at higher salinity (Figure 6). The alkaline phosphatase, *β*-glucosidase, and *N*-acetyl-*β*-D-glucosaminidase activities were lower for all the species, except *S. dacryoidea* for alkaline phosphatase, *R. mucilaginosa* for *β*-glucosidase, and *M. australis* for *N*-acetyl-*β*-D-glucosaminidase (Figure 2 and Figure 3(1)). Also, the hydrolysis of proteins and peptides by leucine-aminopeptidase and trypsin [45,66] was lower in the saline medium for all the studied species (Figure 4). For *S.dacryoidea* and *R. mucilaginosa*, the normalization of these values by the cell abundance allowed us to identify higher activities. As these species had lower cell abundance values, the cell-specific activity increased.

Proteins are tightly associated with water as they tend to bind to the hydrophobic groups in the protein surface [169,170,171]. Water is incorporated inside the protein, which makes it an important part of its native structure, hence its proper function [170,172,173]. Nevertheless, salt ions tend to sequester water into other hydrated ionic structures, which diminishes intermolecular hydrogen bonds and also the availability of free water molecules [174,175]. As stated by Sinha and Khare [176], the salinity effect depends on the salt concentration along with the chemical composition of the protein. Based on our results, we suggest that salinity influenced differently the kinetic parameters V_max_ and K_m_ (Figure 2, Figure 3, Figure 4 and Figure 5). The majority of V_max_ (APA, BGL, BXY, LAP, and TRY) were reduced in the saline conditions, whereas some enzymes (BXY and TRY) exhibited a lower affinity under saline conditions. For the remaining enzymes, there was no detectable clear influence of the salinity on the K_m_. According to Blomberg and Adler [13], under osmotic stress, the capacity for substrate uptake limits growth. This might lead to a decrease in activation energy, and hence, to increase and maximize the reaction rate [177].

Some enzymes, however, are salt-tolerant or halophilic [178]. Compared to other enzymes, the halophilic ones have a surface with more amino acid residues that tend to bind to hydrated cations [179], resulting in a substantially larger multi-layered hydration shell [172]. This reduces the enzyme hydrophobicity, prevents aggregation [172,180], and also allows the halophilic enzyme to adopt a conformational structure that is flexible [176,181,182,183]. In this study, we detected potential halophilic enzymes, which were APA and OLE for *S. dacryoidea*, and SUL for all species except *R. sphaerocarpa* (Figure 3). Interestingly, *R. mucilaginosa* was the only species that did not exhibit a sulfatase activity in the non-saline medium but displayed one in the saline medium. As mentioned above, as sulfate esters are less common in terrestrial plants [184], sulfated polysaccharides might be a physiological adaptation to the high ionic strength of the marine environment [146,147,148,149]. Even though the overall rates of sulfatase were low compared to other enzymes, the studied marine fungi used it, which might be an advantage in marine environments to access monosaccharides from sulfated polysaccharides. Similarly, Gladfelter, James, and Amend [3] and Gonçalves, Hilário [185] supported that as fungal nutritional sources are at high environmental osmolyte concentrations, fungi might have evolved enzymatic activities to compete for its uptake. In addition, we did not find differences in the K_m_ between the saline and non-saline media (Figure 5F). Therefore, the affinity to sulfate esters might depend on the species rather than on salinity.

The results obtained in this study suggest that the salinity effect on the enzyme kinetics of marine fungi depends on the species as well as on the enzyme type, specially for K_m_ (Figure 5). For V_max_, the studied marine fungi seem to be adapted to low and high levels of salinity, but generally, extracellular enzymatic activities were higher in the non-saline medium (Figure 6). Similar results were found for cellulase and xylanase produced in the absence of salts by the mangrove species *Pestalotiopsis sp*. [167]. When these were added to a saline medium, they exhibited lower activities. As suggested by Zaccai [171], each protein seems to have evolved a stable form for a particular environment. This was also proposed for another mangrove fungal species that secreted two enzymes in the presence of salts and another one in the absence of salt [186]. Thus, the release of isoenzymes might be an adaptation mechanism to salinity [167]. As all the studied fungal species displayed enzymatic activities in both media, we suggest that different enzymes might catalyze the same reaction (isoenzymes) and allow them to perform their activities at a wide range of salinities. Nonetheless, it is important to highlight that the degradation of complex compounds also requires a wide range of enzymes [72].

### 4.3. Life Strategies

In the study of Solis, Draeger, and Cruz [187], some marine fungi that released algal degrading enzymes, such as cellulase, were related to the development of ice–ice disease in red seaweed. Other enzymes, such as proteases, were also involved in normal and pathological processes [188]. In terrestrial environments, trypsin-like enzymes are characteristic of plant pathogens [16,23,66,189]. Therefore, the spectrum of secreted enzymes for the hydrolysis of polysaccharides and proteins by some pathogenic fungal strains has been taken as indicative of less specialized nutritional requirements [16]. The present study found that some species are more specialized in cleaving certain substrates, while others appear to be more generalists (Figure 6). *S. dacryoidea* exhibited the highest extracellular enzymatic activity of all the tested fungal strains. Allen, Quayyum [190] found that while the percentage of the total yeast community present on healthy leaves of bentgrass was <9.9%, their biomass increased to 32.0–44.7% on infected leaves. This biomass included yeast species such as *S. dacryoidea*. Therefore, the outstanding combination of cellulase, xylosidase, and trypsin-like activities of *S. dacryoidea* in the non-saline and saline medium suggests less specialized nutritional requirements and potential pathogenicity that could affect plants in terrestrial environments as well as algae in aquatic systems.

Chitinases and trypsin inhibitors have been highlighted as antifungal proteins that are released by the host to prevent or fight against fungal infection [97,191]. In the case of chitinases, fungi can produce them for nutritional, symbiotic, or parasitic purposes [103,111,192], while bacteria can use chitin as a carbon and energy source [193]. Unlike bacteria, many fungi can penetrate solid substrates [2], specially filamentous ones [161], so hyphal growth is an important feature in colonizing substrates [126]. In the present study, *B. parvus* was the only species that had a filamentous structure. Even though the overall rate of NAG was low when compared to other enzymes, *B. parvus* had the highest NAG (Figure 2(3)). Some fungi live on the surface of other organisms as ectobionts, or live within them [2]. Our results suggest that the high release of chitinases by the filamentous species *B. parvus* might be an advantage in penetrating and using this solid substrate. Moreover, the hydrolysis of chitin by the other studied marine fungi might indicate the capability to degrade organic matter of animal origin and other fungi, even if it is performed at low rates.

Some marine fungi species are obligatory marine, whereas others were introduced by river runoff, wind, or other organisms animals [24,131]. Thus, facultative marine fungi are frequent [2]. A comparative study using *Paradendryphiella salina* and its terrestrial relative *Septoria lycopersic* showed that the latter had more genes for cellulose and hemicellulose degradation, but both harbored an equal amount of genes for chitin hydrolysis [84]. Nonetheless, *P. salina* was capable of degrading most components of brown algae, seagrasses, and terrestrial plant material [84]. Hence, this species has a broad and saprobic lifestyle and also colonizes terrestrial and marine environments [84]. *S. dacryoidea* has a worldwide distribution and has been reported from the coast of Portugal [194], New Zeeland [195], and grasslands in Georgia, USA [190], but was originally isolated from the Antarctic Peninsula [54]. This wide distribution, together with our results suggests that the high enzymatic activity of *S. dacryoidea* in both the saline and non-saline media is related to potential facultative marine species that are adapted to different salinities (Figure 6). Moreover, it seems that *S. dacryoidea* conserves enzymes to degrade carbohydrates of terrestrial origin, such as cellulose and xylose, which might be an advantage over obligatory marine species like *M. australis* [53,123].

### 4.4. Potential Environmental Implications

As shown in Figure 1, even though the growth was similar for the majority of species (*R. mucilaginosa*, *R. sphaerocarpa*, and *M. australis*) at 0 g/L and 35 g/L salinities, the array of enzymes secreted was different. Our results suggest that the release of certain enzymes might be favored in freshwater. Similar results were reported by Caruso and Zaccone [30], where higher microbial LAP enzymatic activities were found in the upper layers of marine waters associated with freshwater input. During the second half of the 20th century, more than 80% of the Antarctic Peninsula glaciers have retreated [196], so freshwater input has lowered the salinity of coastal waters [197]. Consistently, salinity variations have been reported in the Arctic [198]. Rojas-Jimenez, Rieck [199] stated that salinity variation influences the fungal community composition. Our results indicate that the studied marine fungal species might be adapted to different salinities. However, their extracellular enzymatic activities were generally higher under freshwater conditions (Figure 6). Hence, in regions where the ocean is freshening, the species with broader enzymatic capabilities at lower salinities might thrive, whereas others might disfavored. The potential stratification of the water column might also influence this activity, as the critical transport of deep-water nutrients might be affected [47,50]. This might lead to the expression of different enzymes, which might influence the role of marine fungi in the biogeochemical cycles of the surface ocean.

## 5. Conclusions

The presented results broaden the current ecological knowledge of marine fungi. Together with similar results on freshwater and terrestrial fungi support the great biochemical versatility of this group of organisms. Generally, the fungal growth was not significantly influenced by the presence or absence of salts. However, using eight fluorogenic substrates as representatives of the main molecules found in the oceans, together with kinetics, allowed us to identify that their extracellular enzymatic activities might be affected differently depending on the species and the enzyme type. Although the studied marine fungi were able to degrade these complex molecules under both conditions, we suggest that this might be generally enhanced in non-saline conditions, with some exceptions like sulfatase. Therefore, we suggest that changes in salinities due to natural and human processes might influence the array of enzymes released by marine fungi and potentially also their role in the biogeochemical cycles.

## Figures and Tables

**Figure 1 jof-10-00152-f001:**
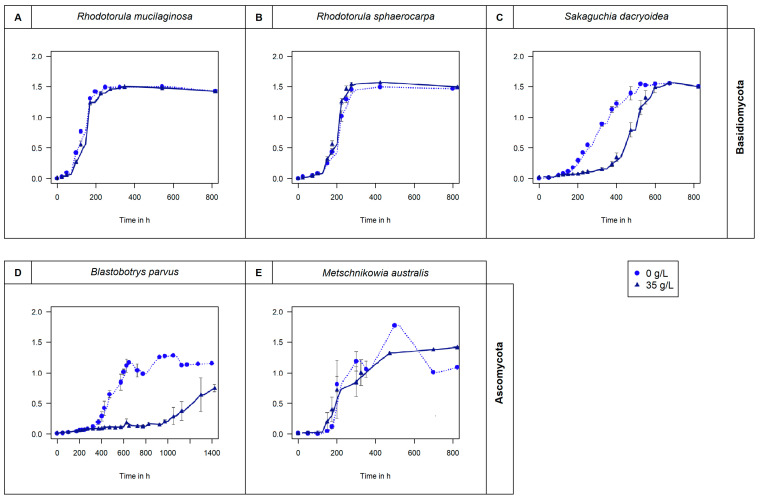
Fungal growth in non-saline (0 g/L) and saline (35 g/L) media measured by optical density (OD_660_) per hour. Basidiomycota species (**A**) *R. mucilaginosa*, (**B**) *R. sphaerocarpa*, and (**C**) *S. Dacryoidea*. Ascomycota species (**D**) *B. parvus*, and (**E**) *M. australis*.

**Figure 2 jof-10-00152-f002:**
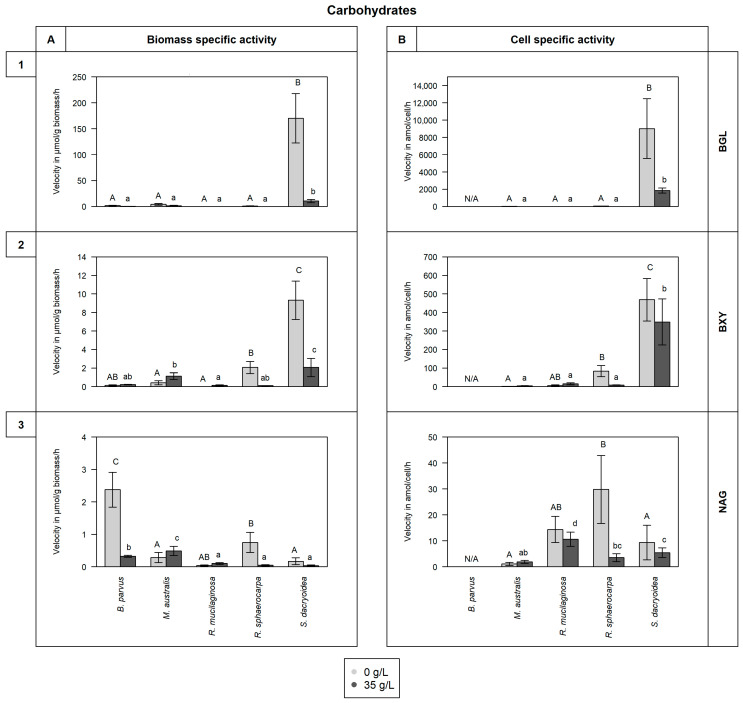
V_max_ in (**A**) μmol/g biomass/h and in (**B**) amol/cell/h obtained from the total enzymatic activity of the fungal strains *B. parvus*, *M. australis*, *R. mucilaginosa*, *R. sphaerocarpa*, and *S. dacryoidea* grown in non-saline (0 g/L) and saline (35 g/L) media. The substrates used represented the hydrolysis of carbohydrates by (**1**) *β*-glucosidase (BGL), (**2**) *β*-xylosidase (BXY), and (**3**) *N*-acetyl-*β*-D-glucosaminidase (NAG). However, due to the filamentous structure of *B. parvus*, its cell abundance was not pssible to calculate. Tukey’s HSD was performed to test for significance (*p* < 0.05) between the extracellular enzymatic activity in the non-saline (light grey and uppercase) and saline (dark grey and lowercase) media, where letters represent the different groups obtained. Bars not sharing any letter (A, a, B, b, C, c, and d) were significantly different, and bars sharing a letter (AB, ab, and bc) were not significantly different.

**Figure 3 jof-10-00152-f003:**
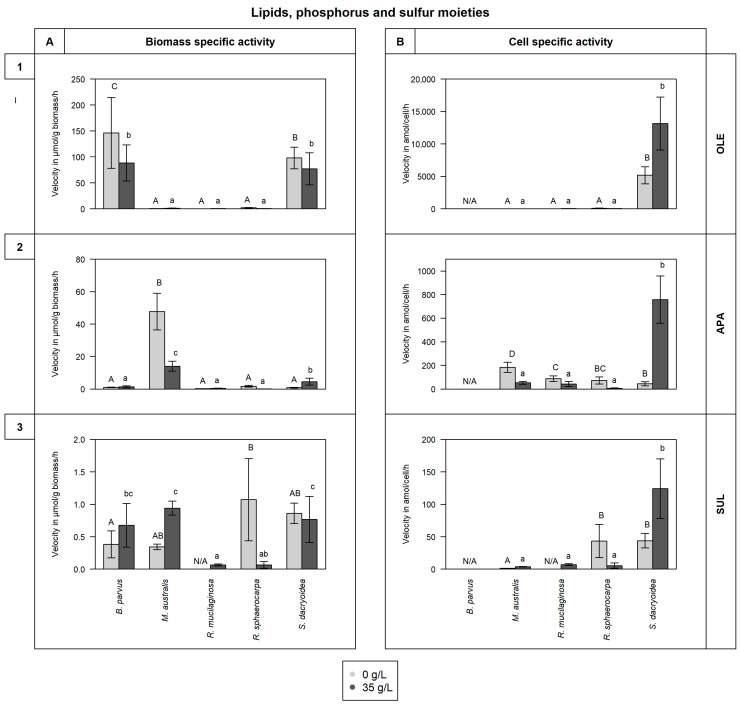
V_max_ in (**A**) μmol/g biomass/h and in (**B**) amol/cell/h obtained from the total enzymatic activity of the fungal strains *B. parvus*, *M. australis*, *R. mucilaginosa*, *R. sphaerocarpa*, and *S. dacryoidea* in non-saline (0 g/L) and saline (35 g/L) media. The substrates used represented the hydrolysis of lipids, phosphorus, and sulfur moieties by (**1**) lipase (OLE), (**2**) alkaline phosphatase (APA), and (**3**) sulfatase (SUL), respectively. However, due to the filamentous structure of *B. parvus*, its cell abundance was not possible to calculate. *R. mucilaginosa* did not exhibit any SUL activity under non-saline conditions, indicated by “N/A”. Tukey’s HSD was performed to test for significance (*p* < 0.05) between the extracellular enzymatic activity in the non-saline (light grey and uppercase) and saline (dark grey and lowercase) media where letters represent the different groups obtained. Bars not sharing any letter (A, a, B, b, C, c, and D) were significantly different, and bars sharing a letter (AB, ab, BC, and bc) were not significantly different.

**Figure 4 jof-10-00152-f004:**
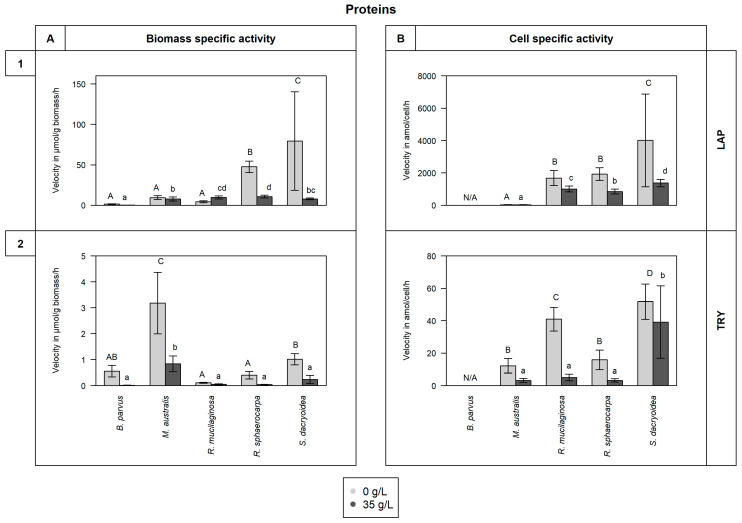
V_max_ in (**A**) μmol/g biomass/h and in (**B**) amol/cell/h obtained from the total enzymatic activity of the fungal strains *B. parvus*, *M. australis*, *R. mucilaginosa*, *R. sphaerocarpa*, and *S. dacryoidea* grown in non-saline (0 g/L) and saline (35 g/L) media. The substrates used represented the hydrolysis of proteins by (**1**) leucine aminopeptidase (LAP) and (**2**) trypsin (TRY). However, due to the filamentous structure of *B. parvus*, its cell abundance was not possible to calculate. Tukey’s HSD was performed to test for significance (*p* < 0.05) between the extracellular enzymatic activity in the non-saline (light grey and uppercase) and saline (dark grey and lowercase) media where letters represent the different groups obtained. Bars not sharing any letter (A, a, B, b, C, c, D, and d) were significantly different, and bars sharing a letter (AB, bc, and cd) were not significantly different.

**Figure 5 jof-10-00152-f005:**
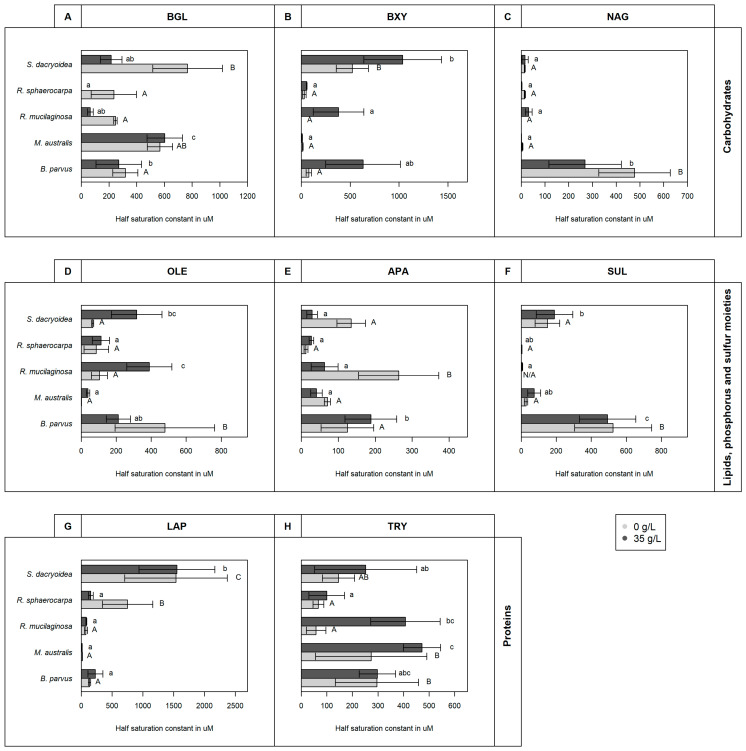
K_m_ in µM obtained from the total enzymatic activity of five marine fungal isolates *B. parvus*, *M. australis*, *R. mucilaginosa*, *R. sphaerocarpa*, and *S. dacryoidea* grown in non-saline (0 g/L) and saline (35 g/L) media, for substrates representatives of carbohydrates: (**A**) *β*-glucosidase (BGL), (**B**) *β*-xylosidase (BXY), and (**C**) *N*-acetyl-*β*-D-glucosaminidase (NAG); lipids, phosphorus and sulfur moieties: (**D**) lipase (OLE), (**E**) alkaline phosphatase (APA), and (**F**) sulfatase (SUL), respectively; and proteins: (**G**) leucine aminopeptidase (LAP), and (**H**) trypsin (TRY). Tukey’s HSD was performed to test for significance (*p* < 0.05) between the extracellular enzymatic activity in the non-saline (light grey and uppercase) and saline (dark grey and lowercase) media where letters represent the different groups obtained. Bars not sharing any letter (A, a, B, b, C, and c) were significantly different (*p* < 0.05), and bars sharing a letter (AB, ab, bc, and abc) were not significantly different.

**Figure 6 jof-10-00152-f006:**
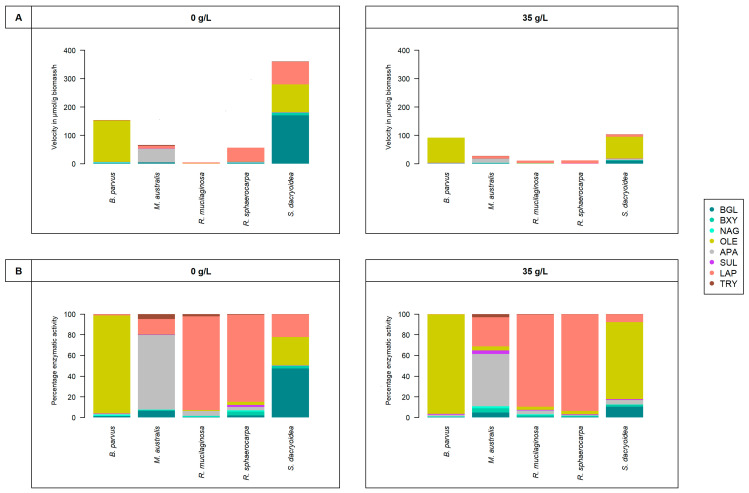
Individual enzymatic contribution of the fungal strains *B. parvus*, *M. australis*, *R. mucilaginosa*, *R. sphaerocarpa*, and *S. dacryoidea* in (**A**) μmol/g biomass/h and in (**B**) percentage for the enzymes *β*-glucosidase (BGL), *β*-xylosidase (BXY), *N*-acetyl-*β*-D-glucosaminidase (NAG), lipase (OLE), alkaline phosphatase (APA), sulfatase (SUL), leucine aminopeptidase (LAP), and trypsin (TRY).

**Table 1 jof-10-00152-t001:** Targeted enzymes with fluorogenic substrate analogues and their respective fluorophore standards (MUF—methylumbelliferyl, MCA—methylcoumarylamide).

Type	Code	Name	Standard
Carbohydrates	BGL	*β*-glucosidase	MUF
	BXY	*β*-xylosidase	MUF
	NAG	*N*-acetyl-*β*-D-glucosaminidase	MUF
Lipids, phosphorus, and sulfur moieties	OLE	Lipase	MUF
	APA	Alkaline phosphatase	MUF
	SUL	Sulfatase	MUF
Proteins	LAP	Leucine aminopeptidase	MCA
	TRY	Trypsin	MCA

## Data Availability

The raw data supporting the conclusions of this article will be made available by the authors without undue reservation to any qualified researcher.

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
