# Peer review of "Influence of Salinity on the Extracellular Enzymatic Activities of Marine Pelagic Fungi"

_jof, 2024, doi:10.3390/jof10020152_

Round 1
Reviewer 1 Report
Comments and Suggestions for Authors
p 3 line 103 “…both containing 2 g/L of glucose, malt extract, peptone, 103 and yeast extract.” The concentration of nutrients and pH should be indicated.
p 3 line 111 “…to achieve an OD660 ≈ 1.” Cultures of filamentous fungi usually grow in the form of pellets. Is it correct to measure biomass using OD660?
p 4 line 155 “This was serially diluted…” What was used to dilute the solution?
p 4 line 163 “To determine fungal biomass…” It is not clear at what point in the culture growth period the biomass was determined?
It is not clear at what pH the enzyme activity was measured? This should have been indicated.
In section 4.2, the authors discuss the effect of salinity on enzymes activity and note that salinity reduces their activity. Is it possible that the decrease in enzymatic activity is not associated with the influence of ionic strength on the protein molecule, but with a decrease in enzyme production by the fungi under salinity conditions?
The work studied the activity of β-glucosidase and β-xylosidase, which are part of a complex of enzymes that decompose xylan and cellulose. To what extent does the study of the activity of one enzyme reflect the activity of the enzymes complex?
p 16 line 568 “ …released algal degrading enzymes such as cellulose.” May be “cellulase”?
Reviewer 2 Report
Comments and Suggestions for Authors
The manuscript “Influence of salinity on the extracellular enzymatic activities of marine pelagic fungi” describes the cultivation of five different fungal strains, two Ascomycetes and three Basidiomycetes, under low and high salinity conditions (0 and 35 g/L NaCl). Apart from the growth curves of the yeast, extracellular enzymatic activities of hydrolases for carbohydrates, lipids and proteins as well as organic phosphorus and sulphur-containing compounds were evaluated.
The manuscript is well written and understandable. The authors collected a multitude of data points for different enzymatic activities and, in general, the data is well presented. Nonetheless, a couple of errors and inconsistencies, especially in the result section, have to be explained or changed. See below for more details.
The material and method section details that the fungal strains were kept at room temperature and at normal salinity for culture maintenance and only set onto high salt conditions with the start of the experiments. Have the authors considered if the fungi might need significantly more time to adjust their metabolism to high salt conditions? This point should at least be mentioned and discussed in the discussion. I would also prefer that the time point of sample for the extracellular activities is mentioned instead of the general ‘once the exponential phase was reached’ (l. 118).
For all figures (but no. 1 and 6), it seems unnecessary that two colours that resemble each other so closely were chosen for the depiction. Please use black and grey or something similar to make life a little easier. If the manuscript is printed in grayscale, the two colours are not discernable right now. For a number of graphs, it might also make sense to divide the y-axis into two parts to enable the authors to discern the height of all those extremely low bars that right now are not distinguishable from the x-axis. I also think that it would be helpful for the readers if, instead of having all graphs and then all the text, the two could be a little more intermingled.
I am not able to understand the indications for the significance testing. Why are some bars labelled with small, some with capital letters? Why and when is it a or b or c? Please add the relevant explanations (apart from a is significant, ab is not) so the readers can follow your testing regime. In addition, some of the significance statements in the main text do not make sense to me. A list:
- Fig. 2.2: How can the cell specific activity be significant in 0, but not at 35 g/L salt for R. mucilaginosa? The same goes for the biomass specific activity of R. sphaerocarpa.
- Fig. 5B, l. 278: Is the km of R. sphaerocarpa really significantly higher at 35 g/L when compared to 0? The same goes for Fig. 5D and R. sphaerocarpa (ll. 303-305).
- Fig. 3.3, l. 324-325: S. dacryoidea also shows lower activity in saline medium.
In some graphs, the highest activity changes between 0 and 35 g/L salt between biomass and cell specific activity. This is the case for fig. 2.3 R. mucilaginosa, fig. 3.1 S. dacryoidea, fig. 3.2 R. mucilaginosa, fig. 3.3 S. dacryoidea and fig 4.1 R. mucilaginosa. The phenomenon is mentioned once in the results (ll. 286-299), but never explained nor discussed why it seems to be specific for those two strains.
A higher km indicates a lower affinity. This correlation was noticed by the authors during the discussion section, but did not stop them from focusing on those fungi and affinity constants that were particularly high, while they should have focused both their results and discussion on particularly low values. Please adjust the text as needed.
Ll. 335-340: The vmax was not higher for R. mucilaginosa in non-saline medium at all. In addition, I am not sure if the difference between both media is significant for M. australis? The second sentence is also not completely true as M. australis shows higher activity than the Basidiomycete R. mucilaginosa.
Ll. 341-342: The km of R. mucilaginosa is lower than the one of B. parvus and the sentence thus not correct. There is also a definitive influence of the salinity for R. sphaerocarpa.
Some additional, minor points:
- Capable is followed by of and gerund: ‘capable of changing’. Please correct where applicable.
- Both is not followed by a comma. E.g.
- The formatting in ll. 125-127 is off.
- Ll. 153: The fluorophore was added to the ‘liquid culture’. Does this mean culture supernatant? Or complete culture? I could not find any information on whether or not the biomass was removed beforehand.
- ß is latin and thus should always be in italics.
- The labelling of the axis in fig. 6 is extremely tiny and hard to read. There should be enough space to increase the font size.
- Ll. 314-315: R. mucilaginosa and R. sphaerocarpa showed higher Vmax than all strains but M. australis. Not than all other strains.
- Ll. 325-326: It should read: The highest vmax in saline medium were detected in S. dacryoidea. And is the activity of M. australis not higher?
- Please delete ll. 359-361.
- Ll. 367-368: Trypsin and the aminopeptidase do not cleave only peptide chains and leucine residues.
- L. 369-371: Could you elucidate what potential bias was supposed to be avoided with the short incubation period?
- Ll. 400-401: The first sentence mentions bacteria as chitin degraders, the second states that they cannot hydrolyse chitin. This seems like a contradiction.
- L. 412: Why should a low vmax indicate a high affinity?
- Ll. 428-429: What about R. sphaerocarpa?
- Ll. 443-445: The sentence reads as if it was better for R. mucilaginosa, but that is not the case.
- L. 445: What is meant with increased flexibility?
- L. 469: Which species does ‘first species’ refer to here? Please clarify?
- Ll. 485-490: The km is actually quite low for M. australis?
- L. 499: A low km was most probably meant. The word ‘low’ is missing.
- L. 578: The sentence is incomplete.
Comments on the Quality of English LanguageThe manuscript is well written and understandable.
Reviewer 3 Report
Comments and Suggestions for Authors
The manuscript entitled "Influence of salinity on the extracellular enzymatic activities of marine pelagic fungi" is certainly of interest, and it is generally well written.
However there are major concerns:
1 - Two Ascomycota e three Basidiomycota. Why? Five Ascomycota and five Basidiomycota would be a better number, including lievitoid and filamentous fungi.
2 - The Authors wished to test the enzymatic actvity in different salinity conditions. However, they used only 2 concentrations: 0 and 35 g/L. Why? At least 2 more points should be added to confirm the Authors' conclusion.
3 - The biotechnological appeal of Marine Fungi finds its reason in their enzymes working efficiently in condition of high salinity. There are many evidence in literature. Maybe the authors should run experiments with more point (see point number 2) including a positive control, finding a fungus whose activity is better at high concentration.
4 - The Authors should better introduce (either in the Introduction or Methods) the concepts of Vmax and Km for enzyme kinetic. In this way the manuscript will be smoother.
5 - Discussion is too long. It would be better to cut it to the essential, with no repetition
6 - The Authors could cut down the number of self citations
7 - Minor comments in the pdf attacched

English is generally fine, I would recommend only a minor check through the manuscript
Round 2
Reviewer 3 Report
Comments and Suggestions for Authors
File attached

English is fine
Author Response
This was corrected.